# Cytotoxic Marine Alkaloid 3,10-Dibromofascaplysin Induces Apoptosis and Synergizes with Cytarabine Resulting in Leukemia Cell Death

**DOI:** 10.3390/md19090489

**Published:** 2021-08-27

**Authors:** Pavel Spirin, Elena Shyrokova, Timofey Lebedev, Elmira Vagapova, Polina Smirnova, Alexey Kantemirov, Sergey A. Dyshlovoy, Gunhild von Amsberg, Maxim Zhidkov, Vladimir Prassolov

**Affiliations:** 1Department of Cancer Cell Biology, Engelhardt Institute of Molecular Biology, Russian Academy of Sciences, Vavilova 32, 119991 Moscow, Russia; elena.j.shirokova@phystech.edu (E.S.); lebedevtd@gmail.com (T.L.); vr.elmira@gmail.com (E.V.); prassolov45@mail.ru (V.P.); 2Center for Precision Genome Editing and Genetic Technologies for Biomedicine, Engelhardt Institute of Molecular Biology, Russian Academy of Sciences, Vavilova 32, 119991 Moscow, Russia; 3Moscow Institute of Physics and Technology (National Research University), Institutskiy Per. 9, 141701 Dolgoprudny, Russia; 4School of Natural Sciences, Far Eastern Federal University, FEFU Campus, Ajax Bay 10, Russky Island, 690922 Vladivostok, Russia; pollianna_95@mail.ru (P.S.); kantemirov_av@dvfu.ru (A.K.); zhidkov.me@dvfu.ru (M.Z.); 5Laboratory of Experimental Oncology, Department of Oncology, Hematology and Bone Marrow Transplantation with Section Pneumology, Hubertus Wald-Tumorzentrum, University Medical Center Hamburg-Eppendorf, Martinistrasse 52, 20251 Hamburg, Germany; s.dyshlovoy@uke.de (S.A.D.); g.von-amsberg@uke.de (G.v.A.); 6Martini-Klinik, Prostate Cancer Center, University Hospital Hamburg-Eppendorf, Martinistrasse 52, 20251 Hamburg, Germany; 7Laboratory of Pharmacology, A.V. Zhirmunsky National Scientific Center of Marine Biology, Far Eastern Branch, Russian Academy of Sciences, Palchevskogo Str. 17, 690041 Vladivostok, Russia

**Keywords:** leukemia, fascaplysin, E2F1, apoptosis, synergism

## Abstract

Myeloid leukemia is a hematologic neoplasia characterized by a clonal proliferation of hematopoietic stem cell progenitors. Patient prognosis varies depending on the subtype of leukemia as well as eligibility for intensive treatment regimens and allogeneic stem cell transplantation. Although significant progress has been made in the therapy of patients including novel targeted treatment approaches, there is still an urgent need to optimize treatment outcome. The most common therapy is based on the use of chemotherapeutics cytarabine and anthrayclines. Here, we studied the effect of the recently synthesized marine alkaloid 3,10-dibromofascaplysin (DBF) in myeloid leukemia cells. Unsubstituted fascaplysin was early found to affect cell cycle via inhibiting CDK4/6, thus we compared the activity of DBF and other brominated derivatives with known CDK4/6 inhibitor palbociclib, which was earlier shown to be a promising candidate to treat leukemia. Unexpectedly, the effect DBF on cell cycle differs from palbociclib. In fact, DBF induced leukemic cells apoptosis and decreased the expression of genes responsible for cancer cell survival. Simultaneously, DBF was found to activate the E2F1 transcription factor. Using bioinformatical approaches we evaluated the possible molecular mechanisms, which may be associated with DBF-induced activation of E2F1. Finally, we found that DBF synergistically increase the cytotoxic effect of cytarabine in different myeloid leukemia cell lines. In conclusion, DBF is a promising drug candidate, which may be used in combinational therapeutics approaches to reduce leukemia cell growth.

## 1. Introduction

Myeloid leukemia is a group of heterogeneous malignant disorders characterized by uncontrolled clonal proliferation of primitive hematopoietic stem or progenitor cells. This disease is divided into two subgroups according to the type of affected blood cell and its chronic or acute forms and include acute myeloid leukemia (AML) and chronic myeloid leukemia (CML). The mechanisms underlying development of myeloid leukemia are still not clearly understood. However, mutations in genes whose proteins are involved in the development of blood precursor cells seems to play a crucial role by causing rearmaments of signaling pathways disturbing the normal hematopoiesis [1,2,3,4]. The chemotherapy with cytarabine (AraC) and anthrayclines is widely used to treat AML [5]. Targeted treatment approaches represent a promising strategy [6]. For example, the introduction to the CML therapy of BCR/ABL inhibitor imatinib raised up to 90% the overall 5-year survival of CML patients. To date, disease persistence and recurrence remain a major challenge in myeloid leukemia therapy [5,7,8]. Potential targets for treating myeloid leukemias comprise receptors of tyrosine kinases [9,10,11,12], proteins contributing to the drug resistance and mutated proteins [13,14], kinases involved in pro-survival signaling pathways [11,13,15,16], components of DNA-reparation system [17,18] as well as kinases involved in cell cycle regulation and autophagy [19,20,21]. Remarkably, CDK4 and CDK6 kinases are frequently overexpressed in leukemia and responsible for G1 to S phase cell cycle transition via retinoblastoma protein (RB) dependent activation of an E2F1 transcription factor. An inhibitor of CDK4/6, palbociclib (PD-0332991), was previously approved by FDA for treatmet of breast cancer and is currently undergoing several clinical trials in other malignancies [22,23,24]. Interestingly, palbociclib has been found to be a promising candidate to enhance AraC-induced AML cell death. Therefore, the inhibitor is assumed to be an attractive therapeutic option for elderly AML patients who are unable to tolerate high-dose AraC therapy [19]. 

Fascaplysin is a red pigment initially isolated from the marine sponge *Fascaplysinopsis* sp. To date, a wide spectrum of biological activities including antifungal, antibacterial, antiprotozoal, and antitumor effects have been reported for this compound [25,26,27,28]. Interestingly, it has been found to affect cell growth of leukemic cells via CDK4/6 inhibition [28] and PI3K/AKT/mTOR signaling pathway disruption [29,30], while in lung cancer cells, fascaplysin-induced cell death is independent from the presence and activity of CDK4, suggesting that other molecular targets are involved [29]. So far, both, pro-apoptotic and pro-survival mechanisms have been reported to be affected by the marine alkaloid. Thus, fascaplysin inhibits VEGFR2, TRKA, survivin, and HIF-1α [29]. In addition, the brominated derivatives, i.e., 3- and 10-bromofascaplysins induce caspase-dependent apoptosis in human leukemia cells at nanomolar concentrations [31]. 3,10-Dibromofascaplysin (DBF) is a novel halogenated fascaplysin alkaloid initially isolated from the marine sponge *Fascaplysinopsis reticulata*, which has been recently synthesized by our group [32]. Treatment with DBF affects cellular metabolism and causes death of prostate cancer cells [32]. In addition, we have reported DBF to inhibit growth of human drug-resistant prostate cancer cells [33] and identified JNK1/2 kinase to be one of the potential targets of this natural compound. Of note, a previously reported functional kinome profiling assay of serine/threonine kinases (STKs) has suggested JNK3, MAPK12/MAPK14, and CDK1/2 to be potentially affected by DBF in cancer cells [33]. However, until now this has not been further investigated. Importantly, cyclin-dependent kinases CDK1/2 are essential for cell division affecting both G1/S and S/G2 cell cycle transitions and therefore are considered to be a promising target for anticancer therapy [34]. Taken together, fascaplysin and its derivatives have a wide spectrum of activity, which seems to vary depending on the cancer type and the drug concentrations. Here, we evaluated the effect of DBF on human leukemia cells in comparison with the established CDK4/6 inhibitor palbociclib, a promising drug candidate for the therapy of leukemia. Additionally, activity of several brominated derivatives of fascaplysin, including an original natural alkaloid fascaplysin, was determined. Finally, we established that combination of DBF with AraC cause pronounced synergistic cytotoxic action in myeloid leukemia cells in vitro.

## 2. Results

### 2.1. 3,10-Dibromofascaplysin (DBF) Suppress Cell Growth of Myeloid Leukemia Cells

First, we measured the cytotoxicity of the marine alkaloid DBF (Figure 1A) in myeloid leukemia cells and compared its cytotoxic activity with the original alkaloid fascaplysin (FAS), two natural previously described brominated derivatives (3BF and 10BF), and recently synthesized mono-brominated new derivatives of fascapysin (2BF) and palbociclib (PD) (Figure 1B). For that we used four human cell lines: K562—chronic myeloid leukemia cell line; MV4;11—acute myeloid leukemia cell line, bearing mutant *FLT3*; U937 cells, widely used as a model for monocytic leukemia study; THP-1—acute monocytic leukemia cells. The cells were counted 72 h post-treatment and the half-maximal inhibitory concentrations (IC50) were determined (Figure 1C). We found that treatment with all studied compounds including DBF and 2BF significantly suppressed cell growth at nanomolar concentrations. Importantly, long-term incubation of cells in presence of DBF significantly suppress the cell grows (Figure 1D). The number of viable cells counted six days post treatment with DBF (800 nM) was 3–4 times less when compared to control cells treated with DMSO (0.016%).

### 2.2. Effect of DBF on Cell Cycle of Leukemia Cells

Previously, fascaplysin derivatives were shown to induce G1 phase arrest of the cell cycle via inhibition of CDK4/6. Hence, we have studied whether DBF may affect cell cycle of leukemic cells. For each cell line, we used the maximum non-toxic and minimum toxic concentrations. Interestingly, the percentage of cells in the G2-population, representing the suppressed transition of G2 phase was significantly increased 72 h after DBF addition. The statistically significant effect on cell cycle progression was observed in all studied cell lines (Figure 2A). 

Interestingly, FAS and its mono-brominated derivatives (3BF and 10BF) were found to affect cell cycle in S phase when added to K562 (Figure 2B). Of note, the addition of PD to MV4;11 and K562 lines expectedly revealed a significantly increased amount of 2N cell fraction, representing PD-induced G1-arrest (Figure 2C). Thus, DBF was shown to affect S/G2 phase of cell cycle when added to all studied cell lines.

### 2.3. DBF Induces Apoptosis of Acute Myeloid Leukemia Cells

Next, the apoptotic effect of DBF on leukemia cells was determined by a double staining assay. A significant increase of Annexin V/PI double-positive (late apoptosis—LA) and Annexin V single-positive cells (early apoptosis—A) following the treatment with DBF (Figure 3A,B), PD (Figure 3C), FAS and its brominated derivatives (2BF, 3BF, 10BF) was observed (Figure 3D), revealing the activation of early and late apoptosis. The most pronounced effect was detected when cells were exposed to the maximum non-toxic concentration of DBF (400 nM for K562 and U937 as well as 800 nM for MV4;11 and THP1) (Figure 3B).

Other brominated fascaplysin derivatives were also found to induce late apoptosis (Figure 3D). Notably, when compared to K562 fascaplysin and its mono-brominated derivatives, they induce necrosis, resulting in an increase of PI-single positive (necrosis, N) MV4;11 cell counts.

### 2.4. Treatment with DBF Induce Alterations in Expression of Genes Responsible for Survival of Leukemia Cells

The obtained data indicate that DBF affects both apoptosis and the cell cycle machinery. Therefore, we evaluated whether DBF regulates genes encoding key proteins involved in cell cycle progression and apoptosis. We used a qRT-PCR technique to determine gene expression of a panel of twenty target genes encoding cyclins, apoptosis regulators, growth factor receptors, and transcription factors involved in leukemia cell growth and proliferation in treated and non-treated cells. Significant alterations in expression of the genes encoding proteins involved in regulation of apoptosis (*BCL2*, *BCL2L1*, and *TP53*) and cell cycle progression (*CCND1*, *CCND2*, *CCNB1*, *CCNA1*, *CCNE1*, and *CDKN1B*) were found (Figure 4A).

Importantly, treatment with DBF significantly affected genes *TP53* and *BCL2* encoding p53 and BCL2 family proteins related to cancer pathophysiology and resistance to conventional chemotherapy [35,36,37,38,39]. We found that treatment with DBF leads to a reduction of *CCNA1* and *CCND2* gene expression encoding Cyclin A1 and Cyclin D2 in all treated cell lines, which partially may explain the anti-proliferative action of DBF (Figure 4A). Furthermore, we measured the expression of genes encoding “master regulators” receptor kinases and transcription factors involved in leukemia development and resistance to chemotherapy. Importantly, the treatment with DBF causes significant changes in the expression of *KIT*, *PDGFRB*, *VEGFR2*, *ABL1*, and *MYC* in all analyzed cell lines (Figure 4A). Five out of twenty studied genes—*CTNNB1* encoding catenin beta 1, a key regulator of WNT signaling; *KIT* encoding tyrosine kinase; as well as *CCNE1*, *NFKB1*, *E2F1* encoding Cyclin E1, *NFκB*, and *E2F1*—were found to be upregulated in three out of four studied cell lines treated with DBF. Three genes *CCND2*, *CCNA1*, and *FLT3* encoding Cyclin D2, Cyclin A1, and tyrosine kinase receptor FLT3 were downregulated in three studied cell lines. To better understand the changes induced by DBF, we analyzed transcriptomic signatures.

Here, DBF induced upregulation of *KIT*, *CTNNB1*, *CCNE1*, *NFKB1*, and *E2F1* expression and downregulation of *CCND2*, *CCNA1*, and *FLT3* expression in at least three out of four cell lines. We used changes in these genes expression to calculate a score associated with DBF-induced gene expression perturbation. Next, we estimated the expression of the eight genes mentioned above in 422 AML samples from the GSE37642 dataset [40]. For each sample the gene expression was normalized to its mean expression in the dataset (Figure 4B). To calculate the score associated with DBF-induced gene expression changes, we summed normalized genes expression considering whether these genes were up- or downregulated by DBF. Then we performed GSEA by comparing transcriptomic data for 20% of AML samples (85 samples) with the highest scores with the rest of the samples (337 samples) (Figure 4C). Most gene ontology (GO) biological processes involved in cell cycle progression were enriched. Six GOs involved in regulating cytokinesis, protein destabilization, and regulation of NOTCH transcription targets were significantly enriched (Figure 4D). Earlier unsubstituted fascaplysin was suggested to be involved in E2F1 regulation via CDK4/6 inhibition [41]. Notably, we found that treatment with DBF significantly increases *E2F1* expression (Figure 4A). E2F1 is a major regulator of cell survival and is involved in both, cell cycle progression and apoptosis regulation [37,38,39,40]. We used the Kaplan–Meier scan from the R2: genomic analysis and visualization platform (http://r2.amc.nl, accessed on 2 August 2021) to find if E2F1 is associated with prognosis (Bohlander (n = 422) AML data set [40]. We found that survival of patients with relatively high expression of E2F1 (n = 178) was significantly higher than patients with low E2F1 expression levels (n = 244) when analyzed the whole data set. Importantly, when the patients were sorted in accordance with FAB classification, we observed a higher survival rate among patients with the most aggressive low differentiated subtypes of AML M0, M1, M2, revealing relatively high E2F1 expression (Figure 4G). 

E2F1 may play a role in a S phase checkpoint, as it was suggested previously by the experiments in which expression of a stabilized form of E2F1 resulted in S phase delay and apoptosis [42]. Several studies suggest the role for E2F1 in a DNA damage checkpoint and apoptosis [43,44,45]. Disbalance in these processes caused by deregulation of S/G2 transition when DNA is under-replicated was shown to cause mitotic catastrophe resulting in cell death [46]. These may partially explain the stimulation of apoptosis and accumulation of cells in S/G2-cell cycle phase caused by DBF treatment (Figure 2 and Figure 3). To evaluate the possible mechanisms, related to E2F1 expression upregulation after treatment with DBF, we used GSEA to identify GO biological processes associated with E2F1 expression in AML cells by comparing the top 20% of AML samples with the highest E2F1 expression with the rest of the samples from the GSE37642 dataset [40]. E2F1 is a common regulating protein involved in both G1/S and S/G2 cell cycle stages transition [47]. Expectedly, we identified 271 enriched GO biological processes, many of which related to cell cycle regulation and cytokinesis (35.7% of all GOs). This is in line with the results obtained from gene expression score analysis in DBF treated cell lines and AML samples (Figure 4E). Next, we grouped all 271 GOs in accordance with biological processes and obtained the pie-chart representing the proportion of gene sets (Figure 4F). Importantly, GSEA analysis evaluated the significant enrichment of pathways involved in regulation of S phase (21% of all evaluated GOs), G2 phase (15%), and M phase (34%) of cell cycle, metabolism (18%), and, furthermore, DNA-reparation machinery and apoptosis (15% of all evaluated GOs) (Figure 4E,F). Taken together, these data suggest that DBF induced apoptosis may be associated with S and G2/M disturbance resulting in apoptosis upregulation. E2F1 is the main actor in this interplay, which suggests its activation may be deregulated.

### 2.5. DBF Induce Activation of E2F1 Transcription Factor in Myeloid Leukemia Cells

To determine the input of DBF on E2F1 activity, we obtained a model cell line with an E2F1 reporter. For that, myeloid leukemia K562 cells were transduced with a lentiviral vector encoding expressing cassette where several E2F1 bind sites (BSs) linked with late adenoviral promoter (lateADEp) drive the expression of marker fluorescent protein mKATE41. mKATE positive cells with expression of E2F1 reporter were sorted and treated with DBF or PD followed by fluorescence intensity analysis (Figure 4A). Next, we measured the sensitivity of the obtained cell line to DBF and PD, representing IC50 values similar to initial K562 cells (Figure 4B). Palbociclib known to inhibit CDK4/6 was used as a reference to verify the obtained reporter system. Expectedly, we found that treatment with PD significantly decreases the mean fluorescence intensity of reporter cells, which signals the suppressed activity of E2F1 (Figure 5C).

Interestingly, when treated with DBF, the opposite effect on the fluorescence intensity in reporter cells was detected, which signals increased activity of E2F1 (Figure 5C). These results suggest that DBF may affect the E2F1 transcription factor when studied in the context of K562 leukemia cells. Of note, it remains unclear whether it acts directly or indirectly on E2F1, which should be further investigated.

### 2.6. DBF Synergistically Acts with AraC to Induce Suppression of Acute Myeloid Leukemia Growth

Cytarabine (AraC) is widely applied in the treatment of leukemia. However, especially high-dose treatment cannot be applied to all patients due to severe side effects, thus the development of approaches to increase its efficiency (and therefore to lower its dose) is of particular interest. To evaluate the clinical relevance of DBF in potential combinational therapies we performed co-treatment with AraC and DBF. Four leukemia cell lines K562, THP-1, MV4;11, and U937 were treated with several combinations of both drugs for 72 h and a dose–response matrix was obtained (Figure 6A).

Next, the synergy distribution plots were obtained and synergy scores were calculated using SynergyFinder 2.0 web application using the ZIP model to capture the drug interaction relationships and the most synergistic areas were detected (Figure 6B). We found that DBF and AraC act synergistically in all studied cell lines. The most synergistic effect was evaluated when THP1 and MV4;11 acute myeloid leukemia cells were treated with combination of these drugs. Importantly, both DBF and AraC significantly reduce the leukemia cell growth when added in combination of non-toxic concentrations (Figure 6C).

## 3. Discussion

Earlier derivatives of fascaplysin were shown to induce cancer cell death via various mechanisms [25,26,28,30,33,48,49]. Furthermore, unsubstituted fascaplysin was demonstrated to affect cell cycle machinery in cancer cells via CDK4/6 inhibition [28]. Here, for the first time, we investigated the anti-leukemic potential of marine alkaloid 3,10-dibromofascaplysin (DBF) alone and in combination with cytarabine (AraC). To better understand the effect of DBF on cell cycle we used well-known CDK4/6 inhibitor palbociclib (PD-0332991) as a reference. Palbociclib was found to be a promising candidate to treat leukemia [19,50,51] and undergoes several clinical trials to treat different malignant disorders [22,52]. Earlier, brominated compounds were shown to affect cell growth, suggesting potential anti-cancer activity [53]. Furthermore, two mono-brominated derivatives of fascaplysin were shown to induce apoptosis in leukemia cells [31]. We found that DBF effectively suppresses the leukemia cell growth in nanomolar concentrations and induces apoptosis. We compared the effect of DBF, original fascaplysin alkaloid, its mono-brominated derivatives, and palbociclib on cell cycle progression and apoptosis. Expectedly, treatment of cells with palbociclib caused appropriate cell cycle phases distribution confirming the induction of G1 phase arrest. In contrast, treatment of myeloid leukemia cells with DBF resulted in accumulation of cells in S/G2 phases. This clearly indicates that action of DBF differs from palbociclib and most likely is not associated with CDK4/6 inhibition. We also found that treatment with DBF induces rearrangement in expression of genes, associated with tumor progression and cell survival mechanisms. Importantly, treatment with DBF causes significant reduction of *CCNA1* and *CCND2* expression (genes, encoding key regulators of cell cycle transition cyclin A1 and cyclin D2 proteins) in most of the studied leukemia cells. Furthermore, we have shown that DBF induces significant alterations in expression of genes encoding tyrosin kinase receptors which are involved in leukemia progression (*KIT*, *VEGFR*, *PDGFR*). This fact suggests that DBF may potentially synergize with tyrosin kinase inhibitors resulting in increased leukemia cell death. This may be of interest, taking into account the resistance of AML patients to therapy with TKI imatinib, which was initially effective in preclinical studies [54,55]. MYC is a transcription factor known to be involved in tumor progression. Overexpression of MYC is associated with poor prognosis of various malignant diseases including leukemia [56,57,58,59]. Interestingly, DBF was found to suppress MYC, when added to K562 or U937 cells. E2F1 is a transcription factor found to mediate both cell proliferation and apoptosis [60]. The disturbance in S/G2 transition and induction of apoptosis was earlier found to be associated with E2F1 deregulation. Here we indicated that relatively high levels of E2F1 expression may be associated with better outcome of myeloid leukemia patients. In line with this, using the E2F1 reporter system we have detected a pronounced E2F1 upregulation in leukemic cells following DBF treatment. Using GSEA, we identified the biological processes associated with high levels of E2F1 in patients with myeloid leukemia. We utilized the data of gene expression profiling in samples with higher E2F1 expression and identified the genes associated with cell cycle progression (*CCNA2*, *CCNE2*, *CCNB1*, *CCNB2* and, importantly, *CCNE1*) which were also found to be upregulated in leukemia cells exposed to DBF. Furthermore, the expression of CDK2 was found to be higher in cells with high level of *E2F1* expression. Previously, we detected that treatment of prostate cancer cells with DBF causes CDK2 activation [33]. CDK2 and its partner cyclin E forms a protein complex which induces phosphorylation of Rb and further E2F1 activation regulating S phase cell cycle progression [61]. In late S phase, E2F1 forms a complex with cyclin A/CDK2 [62], which is essential for S phase transition. Taken together, the data above suggest that treatment of leukemia cells with DBF may inhibit G2/M checkpoint transition. This correlates with cell cycle analysis representing increased amounts of cells in S- and G2 phases and decreased amounts in the G1 phase following treatment with DBF (Figure 2). Importantly, the GSEA analysis of pathways associated with genes whose expression was affected by DBF, revealed an activation of mechanisms involved in cytokinesis (Figure 4D). This may partially explain the apoptosis activation induced by DBF; however, other possible mechanisms involved in suppression of leukemia cell survival under the treatment with DBF cannot be excluded and therefore anticipate the further verification.

Summarizing the described above bioinformatical and experimental data, we suggest that DBF may be used to improve the outcome of patients with relatively lower levels of *E2F1* expression. Finally, we found that DBF significantly increases the cytotoxic activity of cytarabine, emphasizing its potential as a promising candidate for combinational therapy.

Here we found that DBF is a strong inducer of apoptosis of myeloid leukemia cells and suggest that E2F1 may be a target protein, whose activity is modulated by 3,10-Dibromofascaplysin (DBF). Our data indicate that DBF is a promising compound and its anti-leukemia potential should be further investigated.

## 4. Materials and Methods

### 4.1. Reagents and Chemicals

The marine alkaloid 3,10-dibromofascaplysin (DBF) was synthesized and purified as previously reported [32]. 3-Bromofascaplysin and 10-bromofascaplysin were synthesized and purified as previously reported (Zhidkov, M.E. Tetrahedron Lett. 2007) The purity of the compounds has been confirmed using ^1^H NMR and high-resolution mass spectrometry. 2-Bromofascaplysin was prepared with two-step method developed by Zhu et al from tryptamine and 2,5-dibromoacetophenone [63]. 

^1^H NMR (400 MHz, MeOH-d4): *δ* 9.38 (d, *J* = 6.0, 1H), 8.97 (d, *J* = 6.0, 1H), 8.48 (d, *J* = 8.0, 1H), 8.29 (d, *J* = 8.5, 1H), 8.18 (d, *J* = 1.9, 1H), 8.13 (dd, *J1* = 8.5, *J2* = 1.9, 1H), 7.89 (dd, *J1* = *J2* = 8.2, 1H), 7.80 (d, *J* = 8.5, 1H), 7.53 (dd, *J1* = *J2* = 7.5, 1H). ^13^C-NMR (100 MHz, MeOH-d4): *δ* 182.1, 149.1, 147.6, 143.2, 140.6, 136.2, 133.2, 129.7, 128.0, 127.5, 126.1, 125.5, 124.8, 123.5, 121.4, 118.4, 115.5, 114.8. HRMS-ESI, m/z: [M]^+^ calculated for C_18_H_10_^79^BrN_2_O^+^ 348.9974, found 348.9980 (Appendix A).

Palbociclib (PD) (PZ0383, Sigma Aldrich, Taufkirchen, Germany), Cytarabine (AraC) (C1768, Sigma Aldrich, Taufkirchen, Germany).

### 4.2. Cell Lines and Culture Conditions

The human leukemia cancer cell line MV4;11 was purchased from ATCC. K562 and U937 cell lines were kindly provided by Prof. Carol Stocking from the Heinrich-Pette Institute–Leibniz Institute for Experimental Virology. THP-1 cell line was provided by Dr. Nikita Nikiforov from the Institute of Experimental Cardiology, National Medical Research Center of Cardiology. The cell lines used had a passage No. <20 and were continuously kept in culture for a maximum of 4 weeks. Cells were cultured in a humidified atmosphere with 5% CO2 at 37 °C using the RPMI-1640 growth medium supplemented with 2 mM L-glutamine, 100 units/mL penicillin, 100 µg/mL streptomycin, and 1 mM sodium pyruvate and 10% fetal bovine serum (FBS). RPMI-1640, FBS, penicillin/streptomycin, sodium pyruvate, and L-glutamine were purchased from Gibco (ThermoFisher Scientific, Rockford, IL, USA). All cell lines were regularly checked for stable phenotype and mycoplasma contamination.

### 4.3. Cell Viability Assay

The K562, THP-1, MV4;11, and U937 cells were seeded in 48-well plates (5000 cells/well in 200uL/well) and treated with 0–1600 nM of Fascaplysin (FAS), it’s mono-brominated derivatives (2BF, 3BF, 10BF), 3,10-Dibromofascaplysin (DBF), Palbociclib (PD), or Cytarabine (AraC). Fascaplysin and its derivatives were dissolved in 1% DMSO and 1% DMSO was used as a negative control in experiments with Fascaplysin and its derivatives treatment. Cellular viability was assessed 48 h and 72 h post-drug-treatment using 0.4% Trypan blue solution (ThermoFisher Scientific, USA) exclusion in a 1:1 ratio in the Neubauer chamber. Cell viability and IC_50_s were calculated using The GraphPad Prism software v.9.1.1 (GraphPad Software, San Diego, CA, USA).

### 4.4. RNA Extraction and Real-Time PCR

TRIzol reagent (Invitrogen, ThermoFisher Scientific, USA) was used to extract RNA from 5 × 10^5^ cells according to the manufacturer’s protocol. The purity and amount of extracted RNA were determined by NanoDrop ND-1000 (ThermoFisher Scientific, USA) spectrophotometer. An amount of 2 µg of RNA was used for cDNA synthesis by the RevertAid Reverse Transcriptase kit (ThermoFisher Scientific, USA). qPCR was performed using the Maxima SYBR Green Supermix (ThermoFisher Scientific, USA) and CFX96 Real-Time System (Bio-Rad, Hercules, CA, USA). *GAPDH* gene expression was used for normalization. CFX Manager 3.1 software (Bio-Rad, USA) was used to determine Ct values and for expression levels calculation using the delta-delta Ct method. Appendix A represents the sequences of primers used in current study.

### 4.5. Analysis of Cell Cycle Progression

The cells were plated into 6 well plates in concentrations of 8 × 10^5^ cells per well in the final volume of 2 mL and treated with 0–400 nM of FAS, 2BF, 3BF, 10BF, DBF, and PD for 72 h. Then cells were collected in 1.5 mL tubes, washed once with 1 ml of phosphate-buffered saline, and fixed with ice-cold 70% ethanol at 20 °C overnight. The next day, the cells were washed with phosphate-buffered saline, stained with 50 µg/mL propidium iodide (Sigma-Aldrich, Taufkirchen, Germany), and dissolved in 100 µg/mL RNase A (Sigma-Aldrich). All measurements were performed on an LSRFortessa flow cytometer (BD Biosciences, Franklin Lakes, NJ, USA) and analyzed with FlowJo software version 10.0.7 (FlowJo LLC, Ashland, OR, USA).

### 4.6. Analysis of Apoptosis

The cells were plated into 6 well plates in concentrations of 8 × 10^5^ cells per well in the final volume of 2 mL and treated with 0–400 nM of FAS, 2BF, 3BF, 10BF, DBF, and PD for 72 h. Apoptosis was measured by double staining with Annexin V-FITC (Molecular Probes) and propidium iodide (PI) [14]. All measurements were performed on an LSRFortessa flow cytometer (BD Biosciences, USA).

### 4.7. AML Samples Scoring and Gene Set Enrichment Analysis (GSEA)

DBF-induced gene expression scores based on KIT, CTNNB1, CCNE1, NFKB1, E2F1, CCND2, CCNA1, and FLT3 expression were calculated for each sample from the microarray assay GSE37642 [40] according to the following formula:
$$S=\underset{i}{\sum }log2\left(\frac{exp_{i}}{mean_{i}}\right)*C_{i}$$
where *S* is the final score for each sample, *exp_i_* is an expression of one of the eight genes in the sample, *mean_i_* is the mean expression of the gene in the dataset, *C_i_* equals 1 if the gene was upregulated by DBF and −1 if downregulated. Scores were calculated using custom Python 3.8. script and heatmap was generated using the ComplexHeatmap package [64] in Rstudio 1.4. 

GSEA was performed using GSEA 4.1.0 and GO biological processes database v.7.4. The top 20% of AML samples (GSE37642) with the highest DBF-induced gene expression score or E2F1 expression were compared with the rest of the dataset. Sample groups were compared using weighted enrichment with signal-to-noise metrics for gene ranking. For each analysis, 100 permutations were performed. GOs with FDR *q*-value < 0.25 and nominal *p*-value < 0.01 were considered as enriched.

### 4.8. E2F1 Reporter Cell Line Generation and Flow Cytometry

K562 cell line was used for transduction with lentiviral vector bearing expressing cassette where several E2F1 bind sites (BSs) linked with late adenoviral promoter (lateADEp) drive the expression of marker fluorescent protein mKATE. The generation of lentiviral particles and was performed as previously described using HEK293 cells [65]. The pLN379 (FuGW-S(E2F1)p-mKate2) was a gift from Timothy Lu (Addgene plasmid #105177; http://n2t.net/addgene:105177; accessed on 24 August 2021; RRID:Addgene_105177). For transfection, ProFection Mammalian Transfection System (Promega, Madison, WI, USA) was used. At 6 h after transfection, the medium was changed with RPMI containing 20 mM HEPES. After 16 h, supernatants containing viral particles were aspirated and filtered using Millex-GP 0.22 μM filter (Merck KGaA, Darmstadt, Germany). The titers of lentiviral particles in supernatant were measured using HEK293T cells as described before [66]. Titers were consistently higher than 5 × 10^6^ units/mL. For transduction 1 × 10^5^ of K562 cells seeded in a 24-well plate. The transduction efficiency was analyzed by flow cytometry (BD LSRFortessa) followed by sorting of mKATE positive cells by BD FACS Aria III. Six days post sorting, the cells were checked for the presence of mKATE by flow cytometry (BD LSRFortessa). Flow cytometry was used to detect the marker mKATE in transduced cells 48 h after transduction. The cell suspension of transduced cells was centrifuged at 1200 *g* for two minutes and re-suspended in 1× PBS at a concentration of 1 × 10^7^ cells/mL. The samples were analyzed using 488 nm laser for detection eGFP (510/20 filter). Linear forward and side scatter gates were used to eliminate cell clumps and debris. After gating, a minimum of 10,000 events was recorded for each sample. The percentage of mKATE positive cells and mean fluorescence intensity (MFI) were detected. FACS Diva v 5.0 software (BD Biosciences, Heidelberg, Germany) was used for data acquisition and post-acquisition data processing was done with FlowJo X software (Ashland, OR, USA). 

### 4.9. Drug Combination Analysis

The effects (synergistic versus antagonistic) of DBF on the cytotoxic activity of AraC were determined using the Zero interaction potency (ZIP) reference model [67] using the web-based SynergyFinder 2.0 software (https://synergyfinder.fimm.fi, accessed on 24 August 2021). Cell were treated with AraC (0–40 nM for K562, 0–500 nM for THP-1, 0–32 nM for MV4;11, and 0–8 nM for U937) and DBF (0–200 nM for THP-1 and MV4;11, 0–400 nM for K562, and 0–800 nM for U937) alone or in combination. Cells were incubated for 72 h and the viability was measured as described above.

### 4.10. Data and Statistical Analysis

GraphPad Prism software v.9.1.1 (GraphPad Software, San Diego, CA, USA) was used to perform statistical analysis. The figures represent the data shown as mean ± SEM. To compare two groups, we used the unpaired Student’s *t*-test. One-way ANOVA in combination with Dunnet’s test was used to compare the treated groups with the control group (multiple comparisons). All experiments were performed in three replicates, excluding cell cycle and apoptosis analysis performed in four replicates. Statistically significant differences were assumed if *p* < 0.05 and marked by asterisks.

## 5. Conclusions

In summary, marine alkaloid 3,10-Dibromofascaplysin (DBF) exhibits potent anticancer properties in several myeloid leukemia cell lines, which was executed via apoptosis induction and simultaneous inhibition of cell cycle progression in the S and G2/M phase. DBF activates the expression of genes encoding the proteins involved in leukemia cell survival, such as *KIT*, *CTNNB1*, *CCNE1*, *NFKB1*, *E2F1*, and downregulates the expression of *CCND2*, *CCNA1*, and *FLT3* genes. The drug activates transcriptional factor E2F1, which has been suggested as a possible molecular targets of DBF in leukemia cells. Finally, the compound strongly synergizes with cytarabine enhancing its cytotoxicity. Our data indicate DBF to be a promising compound for treatment of human leukemia.

## Figures and Tables

**Figure 1 marinedrugs-19-00489-f001:**
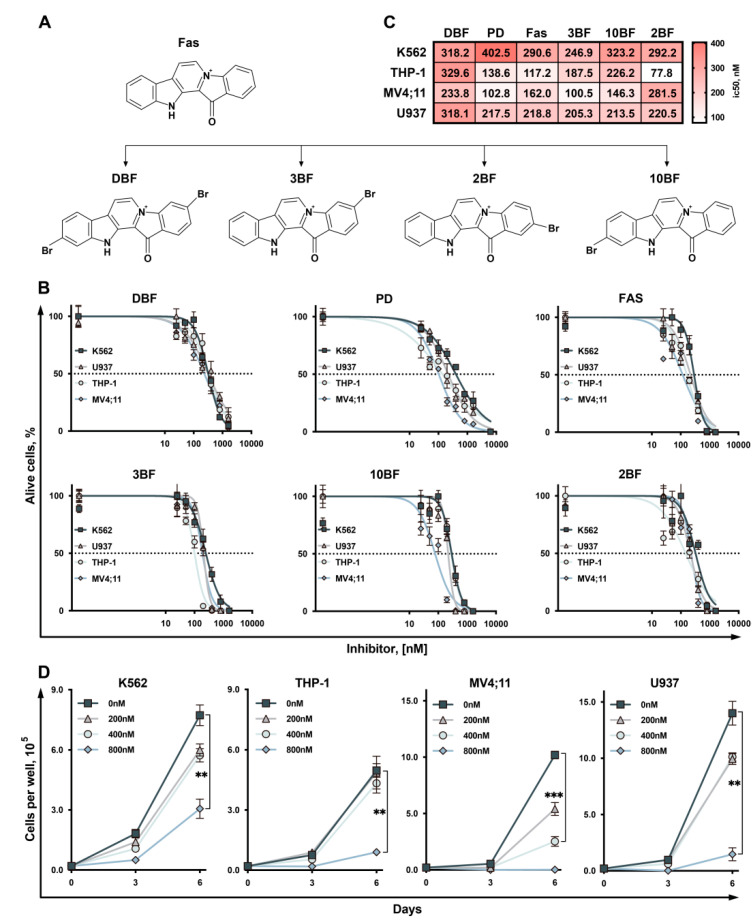
Cytotoxicity of DBF. (**A**) The structure of Fascaplysin (FAS), its mono-brominated derivatives (3BF, 2BF, 10BF), and 3,10-Dibromofascaplysin (DBF). (**B**) Viability of human leukemia cells treated with 0–1600 nM; Dotted line shows the level where percentage of alive cells reaches 50%. (**C**) Heatmap representing IC50 values in treated cells 72 h post-treatment. (**D**) Growth curves represent dose-dependent effect of DBF on cells proliferation after 3 and 6 days. Cell viability was measured 3 days post drug-treatment using trypan-blue exclusion. Asterisks: ** *p* < 0.01; *** *p* < 0.001.

**Figure 2 marinedrugs-19-00489-f002:**
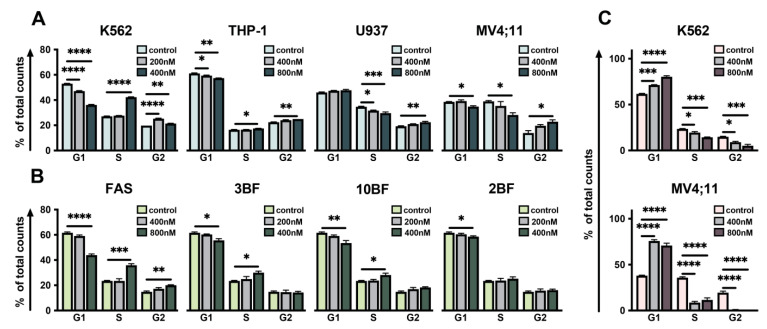
DBF affects cell cycle. Graphs representing the distribution of cells (%) stained with PI (Y axis) among G1, S, G2 cell cycle stages (X axis) performed by flow cytometry (**A**) The effect of DBF on cell cycle of K562, THP-1, U937, and MV4;11 compared to non-treated control (concentrations pointed on legends). (**B**) Effect of Fascaplysin and its mono-brominated derivatives on K562 cell cycle and (**C**) effect of PD on K562 and MV4;11 cell cycle 72 h post-treatment. Asterisks: * *p* < 0.05; ** *p* < 0.01; *** *p* < 0.001; **** *p* < 0.0001.

**Figure 3 marinedrugs-19-00489-f003:**
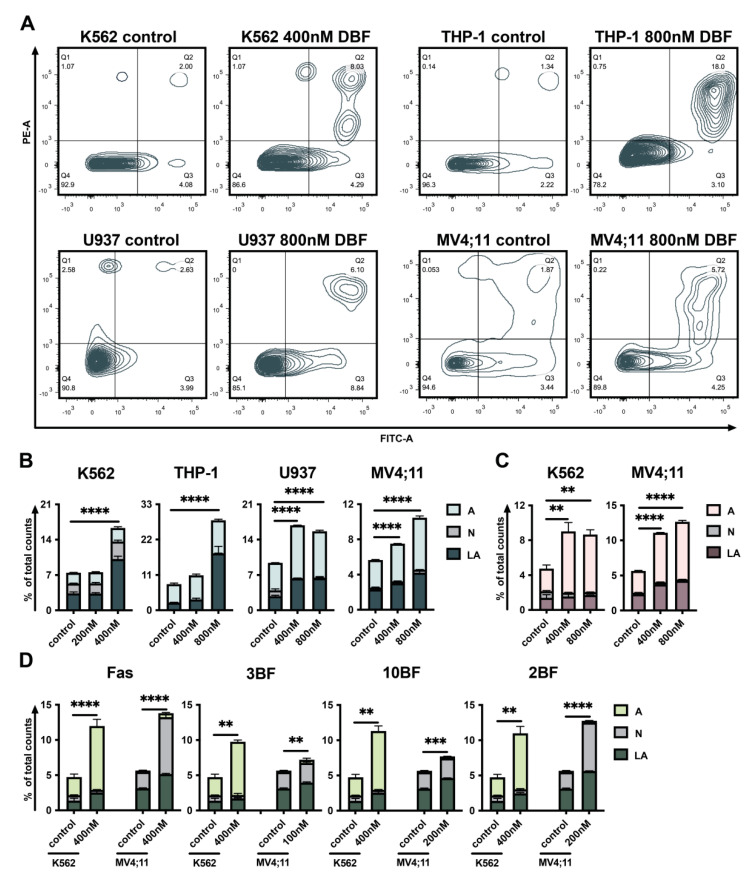
DBF induces apoptosis of leukemia cells. (**A**) The graphs represent distribution of Annexin V/PI stained cells 72 h post-treatment with DBF compared to non-treated control (Q1—PI single positive cells, representing necrosis (N); Q2—AnnexinV/PI: double positive cells, representing late apoptosis (LA); Q3—AnnexinV: single positive cells, representing early apoptosis (A); Q4—AnnexinV/PI: double negative, representing non-stained alive cells. (**B**) The effect of DBF on K562, THP-1, U937, and MV4;11 cells apoptosis. (**C**) The effect of PD on K562 and MV4;11 cells apoptosis 72 h post-treatment. (**D**) Effect of FAS and its mono-brominated derivatives (3BF, 10BF, 2BF) on K562 and MV4;11 cells apoptosis 72h post-treatment. Asterisks: ** *p* < 0.01; *** *p* < 0.001; **** *p* < 0.0001.

**Figure 4 marinedrugs-19-00489-f004:**
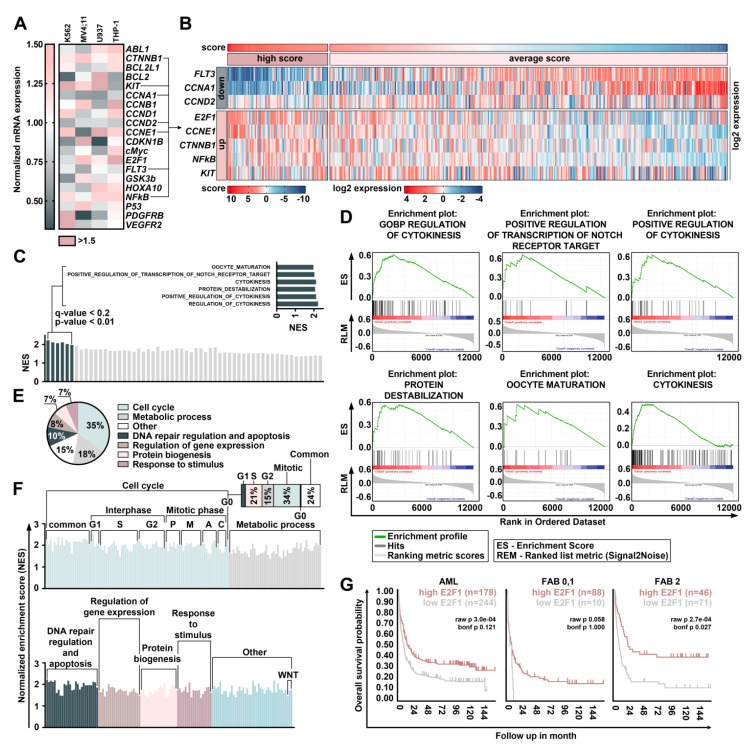
Analysis of genes affected by DBF. (**A**) Heatmap representing relative to GAPDH mRNA expression level of genes) in treated with DBF K562, MV4;11, U937, and THP-1 cells normalized to non-treated controls (fold changes). (**B**) Heatmap representing the clusterization of scores sorted by genes associated with DBF-modulated gene expression and scores in 422 AML samples from the GSE37642 dataset. (**C**) GSEA by comparing transcriptomic data for 20% AML samples (85 samples) with the highest scores with the rest of the samples (337 samples). (**D**) GSEA plots for GOs significantly enriched associated with regulation of cytokinesis, protein destabilization, and Notch transcription targets. (**E**) Pie-chart representing the proportion of GOs associated with the highest E2F1 expression in AML samples. (**F**) The graph representing the grouped proportions of GOs with the highest enrichment scores. (**G**) Kaplan–Meier curves representing the overall survival probability based on the analysis of E2F1 expression in 422 AML samples. Higher E2F1 expression is associated with better outcome.

**Figure 5 marinedrugs-19-00489-f005:**
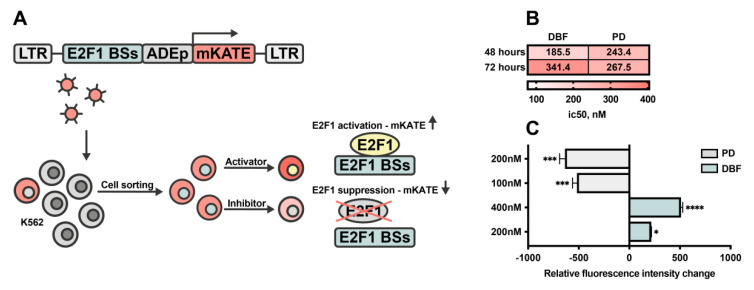
Effect of DBF on E2F1 activity. (**A**) The scheme of E2F1 reporter cell line generation. The expression vector includes: LTR—long terminal repeats, E2F1 BSs—E2F1 binding sites, ADEp—adenoviral early promoter region, mKATE—marker gene encoding red fluorescent protein mKATE. Lentiviral vector encoding expressing cassette was used to obtain lentiviral particles to transduce K562 cells. Cell sorting of transduced cells was performed to obtain mKATE positive cells, bearing E2F1 reporter transgene. The obtained reporter cell line was treated with PD or DBF inhibitors and (**B**) IC50 were evaluated 48 hours and 72 hours post treatment. (**C**) Relative mKATE fluorescence intensity change was measured to analyze E2F1 reporter activity 72 hours post treatment with 100 nM or 200 nM of PD and 200 nM or 400 nM of DBF. Fluorescence intensity was detected using flow cytometry. Asterisks: * *p* < 0.05; *** *p* < 0.001; **** *p* < 0.0001.

**Figure 6 marinedrugs-19-00489-f006:**
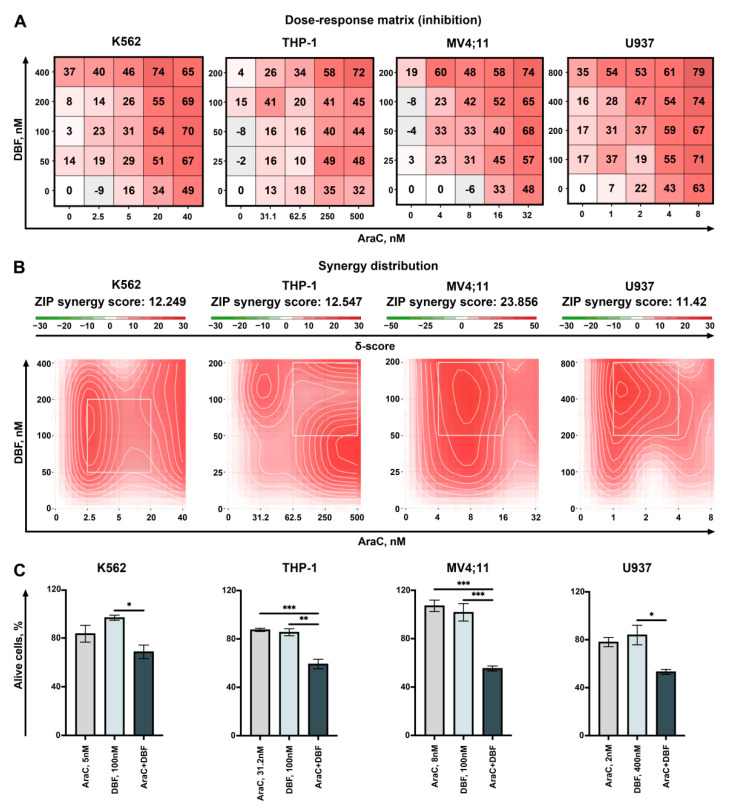
Analysis of the effect on DBF in combination with AraC. K562, THP-1, MV4;11, and U937 cells were co-treated with DBF in combination with AraC. Viability was measured 3 days post drug-treatment using trypan-blue exclusion. (**A**) Dose–response matrix representing the percentage of viable cells compared to non-treated control. (**B**) Synergy plots representing the effect of the drug combination (synergism/additive effect/antagonism) were calculated and visualized using SynergyFinder 2.0 software and a ZIP reference model. Red regions indicate synergism, white—additive effect, green—antagonism. (**C**) The graphs represent the viability of cells in % of control for effective combinations. Asterisks: * *p* < 0.05; ** *p* < 0.01; *** *p* < 0.001.

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
