# Peer review of "Cytotoxic Marine Alkaloid 3,10-Dibromofascaplysin Induces Apoptosis and Synergizes with Cytarabine Resulting in Leukemia Cell Death"

_marinedrugs, 2021, doi:10.3390/md19090489_

Round 1

Reviewer 1 Report

I found the study is sound and results and conclusion are well presented, except for the size of some Figures, which is too small (see my comment below). I addition, I found several typographical errors which should be addressed before the manuscript is accepted.

Page 2 Line 85: italicised Fascaplysinopsis reticulata

Page 3 Line 104: remove full stop after 3,10-

Page 3 Line 108: change syntetized to synthesized

Page 3 Line  115, change 0,016% to 0.016%

Page 3 Figure 1A: the chemical structures for the marine alkaloids are too small, and the atoms in the drawing are too small and are not legible. Please increase the Font size of the atoms in the drawing and also increase the size for figure 1A.

Page 3 Figure 1C: when the numbers in the table is too small to be read in the printed format. Please increase the size.

Page 5 Figure 3 caption Line 155: change non-stined to non-stained

Page 11 Line 359: change discribed to described

Page 10 Line 317: change Palbocliclib to Palbociclib

Author Response

Response to Reviewer 1 Comments

Comments and Suggestions for Authors: I found the study is sound and results and conclusion are well presented, except for the size of some Figures, which is too small (see my comment below). I addition, I found several typographical errors which should be addressed before the manuscript is accepted.

Response to Reviewer

Dear Reviewer,

Thank you very much for your kind revision. We are fully agree with all your suggestions and comments.

Point 1.

Page 2 Line 85: italicised Fascaplysinopsis reticulata

Page 3 Line 104: remove full stop after 3,10-

Page 3 Line 108: change syntetized to synthesized

Page 3 Line  115, change 0,016% to 0.016%

Page 5 Figure 3 caption Line 155: change non-stined to non-stained

Page 11 Line 359: change discribed to described

Page 10 Line 317: change Palbocliclib to Palbociclib

Response 1.

We have corrected the manuscript according to your remarks.

Point 2.

Page 3 Figure 1A: the chemical structures for the marine alkaloids are too small, and the atoms in the drawing are too small and are not legible. Please increase the Font size of the atoms in the drawing and also increase the size for figure 1A.

Page 3 Figure 1C: when the numbers in the table is too small to be read in the printed format. Please increase the size.

Response 2.

We have improved the Figures 1A,C. We increased the size of Figure 1A as well as font sizes of the Figures 1A,C  to make them more legible.

Kind regards,

Pavel Spirin

Reviewer 2 Report

Dear Authors,

Thank you for submitting your valuable manuscript to our journal.

This describes the anti-leukemic effect of marine alkaloid, 3,10-Dibromofascaplysin originated from Fascaplysinopsis sp. 

It has an originality that the compound DBF from marine sponge can be applied as a candidate to the leukemic diseases. 

Overall, there is some points to be fixed technically as well as gramatically.  

Page 2, line 74, Fascaplysinopsis should be written Italic. (same to line 85)

Page 3, line 104, 3,10-. (comma should be removed)

line 108, check the word "syntetized"  (Is it synthesized?)

line 115, (0,016% --> 0.016%)

line 116, Cytotoxity --> Cytotoxicity

Line 127 For evety cell lines, (comma should be added)

cell line "MV4;11" needs to be written in unity - some are written as MV4:11.

(line 149)

line146, Figure(s) 3A, B

The figure legend, especially Fig. 2, Fig. 5, should be written more specific.

Line 219, regulation [37-40].

Line 235, ~treatment with DBF(,)

line 319, This clearly indicate(s)

line 321, ~~DBF induce(s)

line 379, check the formula

Check the expression for cell numbers at line 406,  line 418, line 427 and line 461: 5x105 cells (instead of 5*105), 8x105

line 202-203 and Line 433, check the spaces in the sentence.

Please check other points to be shown as a final version.

Best,

Author Response

Comments and Suggestions for Authors:

Dear Authors,

Thank you for submitting your valuable manuscript to our journal.

This describes the anti-leukemic effect of marine alkaloid, 3,10-Dibromofascaplysin originated from Fascaplysinopsis sp.  It has an originality that the compound DBF from marine sponge can be applied as a candidate to the leukemic diseases. 

Overall, there is some points to be fixed technically as well as grammatically.

Response to Reviewer

Dear Reviewer,

Thank you very much for your suggestions. We have improved our manuscript in accordance to your recommendations. Furthermore we checked the English Language and style and corrected the manuscript.

Point 1.

Page 2, line 74, Fascaplysinopsis should be written Italic. (same to line 85)

Page 3, line 104, 3,10-. (comma should be removed)

line 108, check the word "syntetized"  (Is it synthesized?)

line 115, (0,016% --> 0.016%)

line 116, Cytotoxity --> Cytotoxicity

Line 127 For every cell lines, (comma should be added) cell line "MV4;11" needs to be written in unity - some are written as MV4:11. (line 149)

line146, Figure(s) 3A, B.

Line 219, regulation [37-40].

Line 235, ~treatment with DBF(,)

line 319, This clearly indicate(s)

line 321, ~~DBF induce(s)

Check the expression for cell numbers at line 406,  line 418, line 427 and line 461: 5x105 cells (instead of 5*105), 8x105

line 202-203 and Line 433, check the spaces in the sentence.

Response 1.

We have corrected the manuscript as suggested in your revision.

Point 2

The figure legend, especially Fig. 2, Fig. 5, should be written more specific.

Response 2.

We have improved the Figure legends to make them more detailed. In legend for Figure 2 we included the additional explanations of cell cycle analysis. In Figure 5 legend we added the description of the reporter cell line obtaining procedure and further reporter activity analysis.

Point 3.

line 379, check the formula

Response 3.

We have checked the formula. The given formula is correct. Since fascaplysins are positively charged cations, no loss or gain of atoms or molecules is required for ionization. The obtained value of M+ corresponds to the isomer of this compound of natural origin given in the literature.

Kind regards,

Pavel Spirin

Reviewer 3 Report

Title: Cytotoxic Marine Alkaloid 3, 10-Dibromofascaplysin induces 2 apoptosis and synergises with cytarabine resulting in leukemia cell death

In this manuscript, the authors have studied the effect of the recently synthesised marine alkaloid 3,10-dibromofascaplysin (DBF) in myeloid leukemia cells and found that DBF induced leukemic cells apoptosis and decreased the expression of genes responsible for cancer cell survival. Furthermore, the authors showed that DBF has the capacity to activate the E2F1 transcription factor. Based on the findings, it has been suggested that DEB would be a promising drug candidate for combinational therapeutics approaches.

Overall, this is a well designed and presented study, and it was exciting to read. The introduction and the results are very well written and the quality of the figures is at a high standard. The only minor point here is that the discussion section describes more results repeatedly than discussing the results. This could be improved by linking the results to previous studies showing related molecular-level effects—particularly brominated marine-derived compounds.

In addition, I have a few minor comments on the manuscript.        

Comment 01:

Consider mentioning more details about the used cell lines at the beginning of the result section—For example, K562 - a human erythroleukemic cell line etc.

Comment 02:

Add a figure of 1H NMR or HRMS data to SI to show the purity of the synthesised compounds.  

Author Response

Comments and Suggestions for Authors:

In this manuscript, the authors have studied the effect of the recently synthesised marine alkaloid 3,10-dibromofascaplysin (DBF) in myeloid leukemia cells and found that DBF induced leukemic cells apoptosis and decreased the expression of genes responsible for cancer cell survival. Furthermore, the authors showed that DBF has the capacity to activate the E2F1 transcription factor. Based on the findings, it has been suggested that DEB would be a promising drug candidate for combinational therapeutics approaches. Overall, this is a well designed and presented study, and it was exciting to read. The introduction and the results are very well written and the quality of the figures is at a high standard.

Response to Reviewer.

Dear Reviewer, thanks ever so much for your kind revision. We improved our manuscript in accordance to your recommendations and hope it became more intelligible.

Point 1.

The only minor point here is that the discussion section describes more results repeatedly than discussing the results. This could be improved by linking the results to previous studies showing related molecular-level effects—particularly brominated marine-derived compounds.

Response 1.

The information describing the mechanisms of brominated marine derived compounds action is mostly obscure.  In accordance to your suggestion we included some points describing the application of brominated marine-derived compounds in cancer in discussion section, trying to link it with our results.  

Point 2.

Consider mentioning more details about the used cell lines at the beginning of the result section—For example, K562 - a human erythroleukemic cell line etc.

Response 2.

We added the description of all cell lines used in current study in the beginning of the Results section. 

Point 3.

Add a figure of 1H NMR or HRMS data to SI to show the purity of the synthesised compounds.

Response 3.

We added five Supplementary figures of  1H NMR data, showing the purity of the synthesized compounds used in this study.

Kind regards,

Pavel Spirin